# Genome-Wide Analysis of Tubulin Gene Family in Cassava and Expression of Family Member *FtsZ2-1* during Various Stress

**DOI:** 10.3390/plants10040668

**Published:** 2021-03-31

**Authors:** Shuangbao Li, Peng Cao, Congcong Wang, Jianchun Guo, Yuwei Zang, Kunlin Wu, Fangfang Ran, Liangwang Liu, Dayong Wang, Yi Min

**Affiliations:** 1Department of Biosciences, School of Life and Pharmaceutical Sciences, Hainan University, Haikou 570228, China; 18071008210002@hainanu.edu.cn (S.L.); 17095103210001@hainanu.edu.cn (P.C.); wangcc2278655206@163.com (C.W.); 20071000110013@hainanu.edu.cn (Y.Z.); 19071001210008@hainanu.edu.cn (K.W.); 20071000210028@hainanu.edu.cn (F.R.); 20086000210026@hainanu.edu.cn (L.L.); 2Institute of Tropical Bioscience and Biotechnology, Chinese Academy of Tropical Agricultural Sciences, Haikou 571101, China; guojianchun@itbb.org.cn

**Keywords:** cassava, tubulin, *FtsZ2-1*, duplication event, expression patterns, abiotic stress

## Abstract

Filamentous temperature-sensitive protein Z (Tubulin/FtsZ) family is a group of conserved GTP-binding (guanine nucleotide-binding) proteins, which are closely related to plant tissue development and organ formation as the major component of the cytoskeleton. According to the published genome sequence information of cassava (*Manihot esculenta Crantz*), 23 *tubulin* genes (*MeTubulins*) were identified, which were divided into four main groups based on their type and phylogenetic characteristics. The same grouping generally has the same or similar motif composition and exon–intron structure. Collinear analysis showed that fragment repetition event is the main factor in amplification of cassava *tubulin* superfamily gene. The expression profiles of *MeTubulin* genes in various tissue were analyzed, and it was found that *MeTubulins* were mainly expressed in leaf, petiole, and stem, while *FtsZ2-1* was highly expressed in storage root. The qRT-PCR results of the *FtsZ2-1* gene under hormone and abiotic stresses showed that indole-3-acetic acid (IAA) and gibberellin A3 (GA3) stresses could significantly increase the expression of the *FtsZ2-1* gene, thereby revealing the potential role of *FtsZ2-1* in IAA and GA3 stress-induced responses.

## 1. Introduction

Cell division is the process in which a cell is divided into two cells, which is the basis for the growth, development, and reproduction of organisms. Microtubules are polymerized by tubulins, which participate in cell division as one of the cytoskeleton systems, while tubulins are encoded by multiple gene families in plants [1,2]. The two most typical types of these gene families are α-tubulin and β-tubulin, which account for more than 80% of the amount of tubulin, have similar three-dimensional structures, and can be closely combined into dimers as subunits of microtubule assembly [3,4,5,6,7]. In recent years, γ-tubulin has been discovered, which is located in the microtubule-organizing center and plays an important role in microtubule formation, number and location, polarity determination, and cell division [8,9,10]. These genes are specifically expressed in different tissues and organs, and their mutations may give rise to abnormal plant growth [11]. For instance, the decreased expression of *α-tubulin6* (*TUA6*) gene led to abnormal shoot tip cell division and inhibited root elongation in *Arabidopsis thaliana* [12]. Transgenic rice plants with antisense expression of *β-tubulin8 (OsTUB8*) were inhibited in the amount of seed set after ripening, and the height of plants was 20~60% lower than that of wild type [13].

FtsZ, homologs of tubulin, were first found in bacteria and involved in the bacterial division as cytoskeletal proteins, which became key proteins of chloroplast division in higher plants with the occurrence of endosymbiotic events [14,15,16,17,18,19]. Generally, unlike bacteria just only one *FtsZ* gene, most plants contain three functionally complementary genes, *FtsZ1-1*, *FtsZ2-1*, and *FtsZ2-2*, which presumably arose by gene replication of a single ancestral *FtsZ* gene [20,21,22,23,24,25]. In plants, *FtsZ1-1* or *FtsZ2-1* null mutants can cause chloroplast division defects. Compared with the wild type, a decreased number and increased size of chloroplasts in the cell of green leaf were observed. Unlike *FtsZ1-1* and *FtsZ2-1* mutants, the phenotype of the *FtsZ2-2* null mutant is milder, and the number and size of the chloroplast are less variable [26,27]. During leaf growth, with the increase of chloroplasts in mesophyll cells to maximize photosynthesis, there is a negative correlation between chloroplast size and photosynthetic N-use efficiency (PNUE) [28,29].

The cytoskeleton tubulin and FtsZ proteins have similar functions; both are involved in polymer formation and play a key role in cell division, while tubulin and FtsZ proteins have little sequence identity but showed high conformational similarity [16,30]. Tubulin and FtsZ are GTPases, and their nucleotide-binding sites of tubulin and FtsZ are similar to those of GAPDH (glyceraldehyde-3-phosphate dehydrogenase) but different from nucleotide-binding sites of other typical GTPases [31,32]. Therefore, tubulin and FtsZ form a unique family of cytoskeletal GTPases [33]. Moreover, previous studies suggest that tubulin may have evolved from FtsZ at the beginning of eukaryotic evolution, which leads to extreme sequence divergence related to the shift in function [34].

Cassava is an important cash crop and food source growing in Africa, Asia, and tropical America because of its high starch content in storage root; about 750 million people depend on cassava as food [35,36]. Meanwhile, cassava can also be processed into flour, starch, animal feed, and alcohol [37]. It has strong tolerance and survival ability under biotic and abiotic stresses, and a relatively high yield under poor soil conditions [38]. This makes the production benefit of cassava significantly higher than other crops and more conducive to the rational allocation and utilization of land resources. Drought and salt are the main abiotic stresses affecting the growth of cassava [39]. In many coastal areas, the saline–alkali content in soil is often too high, causing harm to plants, while appropriate salt content can improve the nutritional level and starch content of cassava leaves. [40]. Spraying exogenous hormones can significantly affect the growth and development of cassava; for instance, a high concentration of MeJA could induce the cassava defense mechanism to play a role in advance [41,42,43]. The application of auxin can stimulate the growth of cassava stem and bud, and tubulins play an important role in this process [44,45,46]. Tubulins showed different responses under various stresses, such as high expression of soybean *β-tubulin1* gene under high concentration of auxin stress, while good stability under other abiotic stresses; the expression of *α-tubulin2* gene from *Hevea brasiliensis* was regulated by NaCl, drought, and MeJA [47,48]. Based on the importance of *the tubulin* gene family during cassava growth, especially the regulation of FtsZ on chloroplast, it is of great significance to research the function of the cassava *tubulin* gene family to improve the quality of cassava. For this reason, we analyzed the gene characteristics, phylogeny, gene structure, gene repetition, protein motif, and expression profiles in various tissues, of cassava tubulin family members, and the expression of the *FtsZ2-1* gene was studied under IAA, MeJA, ABA, GA3, salt, and drought treatments.

## 2. Results

### 2.1. Identification of the Tubulin Proteins in Cassava

Through BLAST (basic local alignment search tool) search and HMMER (biosequence analysis using profile hidden Markov Models) analysis, 23 genes with tubulin domain were identified and annotated as MeTubulins in the cassava genome. According to the order of screening, 23 *MeTubulin* genes were named from *MeTubulin1* to *MeTubulin23*. Table 1, Appendix A described information of the 23 MeTubulins, including gene ID, size, molecular weight (MW), isoelectric point (pI), and predicted subcellular location, chromosome number, the total number of positively and negatively charged residues, and sequence. MeTubulin20 and MeTubulin23 were identified as the smallest proteins, both with 421 amino acids (aa), while the largest protein was MeTubulin10 (493 aa). In length with an average of 457 aa, the MW varied from 43.3 kDa (MeTubulin20) to 53.4 kDa (MeTubulin6), and the pI ranged from 4.68 (MeTubulin17) to 6.93 (MeTubulin20). The predicted subcellular localization indicated that MeTubulin10 and MeTublin20 were positioned in chloroplast stroma, MeTubulin2 was shown to be localized in the chloroplast thylakoid membrane, and the rest of them were positioned in the cytoplasm.

### 2.2. Multiple Sequence Alignment, Phylogenetic Analysis, and Classification of MeTubulins

As shown in Figure 1, using the tubulin protein sequences of cassava, *Oryza sativa*, *Hevea brasiliensis*, *Arabidopsis thaliana*, *Dioscorea rotundata*, and *Vitis vinifera*, the unrooted phylogenetic trees were constructed by the maximum likelihood (ML) method, and their evolutionary relationship were further analyzed. A total of 115 tubulins contained 31 *Hevea brasiliensis*, 16 *Arabidopsis thaliana*, 16 *Oryza sativa Japonica*, 11 *Dioscorea rotundata*, and 18 *Vitis vinifera*, which were divided into groups I, II, III, and IV corresponding to β-tubulin, α-tubulin, γ-tubulin and FtsZ based on the four types of tubulin (Appendix A). In addition, referring to the classification of Radchuk et al. [49], group I can be divided into three subgroups, Ia to Ic, and group II can be divided into two subgroups, IIa and IIb. Group Ia consisted of 22 tubulins, usually containing two introns, including five MeTubulins (MeTubulin7, 12, 13, 17, and 19), together with six HbTubulins from *Hevea brasiliensis*, four AtTubulins from *Arabidopsis thaliana*, four OsTubulins from *Oryza sativa Japonica*, three VitTubulins from *Vitis vinifera*, and one *DrTubulin* from *Dioscorea rotundata*. Three MeTubulins (MeTubulin1, 3, and 5), one AtTubulin, three VitTubulins, three OsTubulin10, two HbTubulins, and one DrTubulins belonged to group Ib, which generally contains an extra intron in the 5′ untranslated region. Group Ic mainly consisted of dicotyledons, four MeTubulins (MeTubulin4, 9, 14, and 18), two AtTubulins, seven HbTubulins, and three ViTubulins, and only two monocotyledons OsTubulin10, and DrTubulin2. Similarly, group IIa usually contains three introns, including three MeTubulins (MeTubulin8, 22, and 23), together with two AtTubulins, four HbTubulins, two OsTubulins, four VitTubulins, and two DrTubulins. Group IIb generally contains four introns, consisting of three MeTubulins (MeTubulin11, 15, and 21), two AtTubulins, two VitTubulins, two OsTubulins, four HbTubulins, and two DrTubulins. Two MeTubulins (MeTubulin 6 and 16) clustered with two AtTubulins, DrTubulin7, three HbTubulins, OsTubulin4, and ViTubulin13 in group III. Group IV consisted of three MeTubulins (MeTubulin 2, 10, and 20), three AtTubulins, three DrTubulins, three OsTubulins, two VitTubulins, and three HbTubulins. HbTubulin4 was excluded from the grouping because it was too divergent. Furthermore, a total of four sister pairs of MeTubulins were showed in the phylogenetic tree, including MeTubulin2–MeTubulin10, MeTubulin8–MeTubulin23, MeTubulin15–MeTubulin21, MeTubulin12–MeTubulin19. Tubulin domain sequence alignment of MeTubulin proteins showed that sequence varied greatly from position 33 to 46 amino acids, but the sequence was identical or similar in the same group (Figure 2).

### 2.3. Gene Structure and Motif Composition of MeTubulins

For this study, 10 conserved motifs of MeTubulins with a length of 20 amino acids were identified by the MEME program; the function of these motifs have not been clarified (Appendix A). As exhibited in Figure 3A,B, it was found that MeTubulins with similar motif composition tend to be clustered together. For instance, groups I and II contain all motifs (except for Metubulin23), group III mainly lacks motifs 3, 7, and 2, and group IV only shares a common motif 9.

In order to gain a better understanding of the evolution of the cassava tubulin family, the exon–intron structures of all identified *MeTubulin* genes were analyzed. All *MeTubulin* genes displayed 3 to 11 exons (12 with 3 exons, 3 with 4 exons, 3 with 5 exons, 1 with 6 exons, 2 with 7 exons, and 2 with 11 exons) in Figure 3C. Significantly, each group has the same number of exons (except for the IV group, two contain seven exons and one contains six exons). Meanwhile, the number of introns in the same group is also similar. Generally, the diversity of exon–intron structure makes the gene family show a variety of different functions and could be used as a basis for phylogenetic grouping.

### 2.4. Chromosome Distribution and Synteny Analysis of MeTubulins Gene

In total, 23 *MeTubulins* were unevenly distributed on 11 chromosomes, and each chromosome contained 1-3 *MeTubulins* in Appendix A. A total of 11 segmental duplication genes were obtained from 23 *MeTubulins* by BLASTP and multiple collinearity scan toolkit (MCScanX) methods in Figure 4 (Appendix A). The results show that these *MeTubulin* genes might be produced by segmental replication events.

To research the evolutionary mechanisms of the cassava *tubulin* family further, five comparative syntenic maps of cassava associated with five representative species were constructed, including two monocots (*Oryza sativa* and *Dioscorea rotundata*) and three dicots (*Populus trichocarpa*, *Arabidopsis thaliana*, and *Vitis vinifera*) (Figure 5). In general, cassava and dicotyledons have more collinear genes than monocotyledons (Appendix A). For example, *Dioscorea rotundata* (7), and *Oryza sativa* (1) have fewer collinear genes than *Vitis vinifera* (18), *Populus trichocarpa* (14), and *Arabidopsis thaliana* (9). Some MeTubulins have been observed to be associated with more than one syntenic gene pairs, especially cassava and *Vitis vinifera*, such as MeTubulin9 and MeTubulin17, speculated that these genes might play a significant role in the evolution of the MeTubulin family.

### 2.5. Expression Profiles of MeTubulins in Different Tissue Types

The RNA-seq data publicly available from the GEO (gene expression omnibus) database under the accession number GSE7951 [50], which contained expression profiling (Appendix A) of 11 cassava tissue types (root apical meristem (RAM), leaf, petiole, stem, midvein, lateral bud, storage root, fibrous root, stem apical meristem (SAM), in addition to organized embryogenic structures (OES) and friable embryogenic callus (FEC) for genome editing and transgene integration) were used in expression analysis of cassava *Tubulin* genes (Figure 6). We found that some *MeTubulins* have similar expressions in the same tissue. *MeTubulin1*, *MeTubulin3*, *MeTubulin4*, *MeTubulin5*, *MeTubulin6*, *MeTubulin10*, *MeTubulin17*, *MeTubulin19*, *MeTubulin22*, and *MeTubulin23* showed high levels of expression in petioles and stems, of which *MeTubulin10* was also highly expressed in storage root. *MeTubulin2*, *MeTubulin7*, *MeTubulin9*, *MeTubulin11*, *MeTubulin12*, *MeTubulin13*, *MeTubulin15*, *MeTubulin20*, and *MeTubulin21* were highly expressed in leaves and midveins, *MeTubulin14*, *MeTubulin16*, and *MeTubulin18* exhibited high expression levels FEC and SAM, and *MeTubulin8* exhibited higher expression levels in RAM and storage roots.

### 2.6. Expression Patterns of FtsZ2-1 Gene in Response to Various Stresses

Cassava seedlings were treated with IAA, MeJA (methyl jasmonate), ABA (abscisic acid), GA3, NaCl, and PEG (polyethylene glycol) treatments to research the expression of the *FtsZ2-1* gene by different abiotic stresses and hormonal treatments (Figure 7). Under 100 μM IAA treatment, the expression of *FtsZ2-1* fluctuated with time. The expression of *FtsZ2-1* increased at the beginning, then decreased and subsequently increased again, reached the peak at 8 h, and decreased sharply at 12 h. After that, the expression level approached the peak at 48 h. After treatment with 100 μM MeJA, the expression of *FtsZ2-1* fluctuated with time; it decreased significantly after treatment, reaching the lowest value at 4 h, and then increased to the highest value at 24 h. Under 100 μM GA3 treatment, the expression of *FtsZ2-1* showed an upward trend with the delay of time, decreased at 8 h, and then continued to rise for 48 h to reach the peak. Under the abiotic stress of 300 mM NaCl and 20% PEG, the transcriptional level of *FtsZ2-1* decreased at first and then increased. The difference was that the expression level of *FtsZ2-1* reached the lowest level at 4 h and reached the highest level at 48 h after the treatment with 300 mM NaCl, which was significantly higher than that in the control group. However, under 20% PEG treatment, the transcriptional level decreased rapidly to the lowest level at 2 h and increased to a similar level as that of the control group at 48 h.

## 3. Discussion

Tubulins consist of cytoskeleton proteins that are present in all plant species. Understanding the role of tubulins in growth and non-specific response to abiotic stress factors in plants is essential [6,51]. In our study, the phylogenetic tree, genetic structure, gene replication protein motif, and expression profiles in different tissues of the tubulin family in cassava were analyzed. Furthermore, the expression of the *FtsZ2-1* gene under various stresses was researched. *Tubulin* genes were screened from the cassava genome, along with six other species, for instance, *Arabidopsis thaliana* [52], *Dioscorea rotundata* [53], *Hevea brasiliensis* [54], *Oryza sativa* [55], and *Vitis vinifera* [56]. In this study, 23 cassava *tubulins* were found and renamed as *MeTubulin1* to *MeTubulin23* in the order of screening, lower than *Hevea brasiliensis* (31), but higher than *Arabidopsis thaliana* (16), *Dioscorea rotundata* (11), *Oryza sativa* (16), and *Vitis vinifera* (18); according to amino acid sequences, motif composition, intron structure, phylogenetic feature, and tubulin type of cassava, the 23 cassava *tubulin* genes can be divided into four main groups.

Multiple sequence comparisons were used to analyze the conserved domains of cassava tubulins. The results showed that *MeTubulin23* had the least conserved residues, and *MeTubulin2*, *MeTubulin10*, and *MeTubulin23* had low similarity with other amino acid sequences (Appendix A). The gain and loss of domain is the diverging force of gene family members’ diversity. The sequence variation of domains is relatively common in rice and maize [57,58,59]. Thus, the function and binding specificity of specified four tubulin proteins deserve further study.

The phylogenetic tree divides tubulin proteins into four main groups, and each group contains only one tubulin type from cassava, *Vitis vinifera*, *Oryza sativa*, *Hevea brasiliensis*, *Dioscorea rotundata*, and *Arabidopsis thaliana*, and the neighboring tubulins in the same group might have similar functions; for instance, *HbTubulin27*, *VitTubulin6*, *AtTubulin1*, and *OsTubulin3* are involved in the regulation of plastid division [27,60,61,62]. Overall, 10 conserved motifs were identified in the cassava tubulin family, and it was found that members with similar motif composition had cognate gene structures and were clustered together on the phylogenetic tree. This indicates that the tubulin family is evolutionarily diverse and conservative.

In the course of evolution, gene diversity proceeds through sequence divergence, recombination, and replication [63]. Gene replication events have been often considered important sources of evolutionary dynamics [64]. Tandem duplication and segmental replication lead to the expansion of gene families [65]. Based on the explanation of Holub, tandem duplication of a gene is a chromosomal region within 200 kb containing two or more genes [66]. In the MeTubulin family, seven pairs of genes were found to be derived from fragment repeat evolution, but no tandem repeat was found, which means that the driving force of the family expansion is mainly fragment repeat (Figure 4). Purification selection was the main driving force for differentiation of MeTubulin replication based on Ka/Ks values below 1 (Appendix A) [67]. The *FtsZ2-1* gene in cassava has fragment duplication and a colinear relationship with corresponding genes in dicotyledons (*Arabidopsis thaliana* and *Vitis vinifera*) and monocotyledon (*Dioscorea rotundata*).

The expression profiles of *MeTubulin* in various tissues were analyzed by using RNA-seq (RNA sequencing) data of cassava in the GEO database, which is helpful to research the potential function of the *MeTubulin* gene (Figure 7), and several *MeTubulins* showed tissue-specific expression in specific tissues. For instance, *MeTubulin2*, *MeTubulin10*, and *MeTubulin20* are highly expressed in leaf, and they are involved in the formation of chloroplast in mesophyll cell, which is similar to the homologous genes *AtTubulin8*, *AtTubulin15*, and *AtTubulin1* in *Arabidopsis thaliana* [68,69]. Hence, highly expressed genes might play a role in the physiological process of corresponding tissues, which provides a new idea for researching the potential function of *MeTubulin* genes in cassava.

To date, accumulated pieces of evidence show that plant hormones and abiotic stress play a crucial role in the growth and development of cassava [70,71]. *FtsZ2-1*, a member of the tubulin family, was selected for different stress treatments, and its function was further explored through the transcriptional response to different treatments in our research. IAA, which may be produced by the terminal buds, limits water transport to the lateral buds, and the change of water movement alters the composition of membrane lipids, thus inducing the growth of lateral buds [72]. Exogenous MeJA could delay cell deterioration, maintain postharvest longevity, reduce cell oxidative damage, and regulate storage quality by activating superoxide dismutase, catalase, and peroxidase [70]. In addition, it can induce plant chemical defense by stimulating the expression of plant defense genes [73]. ABA plays a regulatory role in plant growth and under the influence of the external environment [74]. The application of ABA, through foreign aid, can accelerate the adaptation of some crops to cold and drought [75,76]. Furthermore, it was found that GA3 affected the synthesis of cassava starch by regulating the activities of key enzymes in the process of starch synthesis [77]. It was noteworthy that the expression pattern of the *FtsZ2-1* gene is not completely consistent with that of previous studies, which may be caused by the inhibition of growth by a high concentration of IAA and the promotion of growth at low concentration, indicating that *FtsZ2-1* is closely related to the growth of cassava. [67]. The expression of *FtsZ2-1* changed significantly under MeJA treatment, which provides a new idea for studying the functional mechanism of MeJA in cassava. However, there was little difference in the expression of *FtsZ2-1* under ABA treatment, which indicated that *FtsZ2-1* may not be an important factor affecting the mechanism of action of ABA. The level of *FtsZ2-1* transcripts increased significantly after GA3 treatment, indicating that *FtsZ2-1* might be an important member involved in the regulation mechanism of GA3. The main abiotic environmental stresses are salt and drought stresses. Solving water shortage and salt stress is a global problem to ensure the survival of agricultural crops and sustainable metabolism of food production [78]. The expression trends of the *FtsZ2-1* gene induced by PEG and NaCl were similar, which provides an opportunity to study the relationship between drought and salt action mechanism. From these results, the *FtsZ2-1* gene might play a role in cassava growth and environmental stress.

## 4. Materials and Methods

### 4.1. Identification of the Tubulin Gene Family

The whole genome sequence of cassava was downloaded from the EnsemblPlant database (http://plants.ensembl.org/Manihot_esculenta/Info/Index, access date: 22 June 2020). Local BLAST searches were performed based on the hidden Markov model (HMM) profile of tubulin domains from the Pfam database under the accession number PF00091 (http://pfam.xfam.org/family/PF00091#tabview=tab6, access date: 1 July 2020) [43,79]. The selected tubulin protein sequence was submitted to the CDD (conserved domain database) website (https://www.ncbi.nlm.nih.gov/cdd/, access date: 12 July 2020) and SMART website (http://smart.embl.de/, access date: 8 July 2020) with default parameters to confirm the conservative tubulin domain [80]. The sequence length, MW, pI, and subcellular location prediction of cassava tubulin proteins were obtained by online ExPASy (http://web.expasy.org/protparam/, access date: 5 August 2020) and Psort server (http://psort1.hgc.jp/form.html, access date: 1 August 2020) with default parameters.

### 4.2. Sequence Analysis and Structural Characterization

The gene structures were analyzed by the gene structure display server (GSDS: http://gsds.cbi.pku.edu.cn, access date: 21 August 2020) [81]. The structure of exon/intron was determined by comparing genomic DNA and CDS sequences of tubulin genes. The MEME online program (http://meme.nbcr.net/meme/intro.html, access date: 20 September 2020) was used to analyze the conserved motifs in full-length tubulin proteins, with the following parameters: the site distribution, any number of repetitions; the number of motifs, 10; optimum motif length = 10–20 residues, and shuffle the sequences several times under the same parameters to ensure the reliability of motif [82]. Meanwhile, all identified motifs were annotated according to InterProScan (http://www.ebi.ac.uk/Tools/pfa/iprscan/, access date: 8 March 2021).

### 4.3. Chromosomal Localization and Gene Duplication

The location of *MeTubulin* genes on the chromosome was analyzed by MapChart software (https://www.wur.nl/en/show/Mapchart.htm, access date: 20 October 2020) [83]. Multiple collinearity scan toolkit (MCScanX) was used to analyze the gene duplication events, and the parameters were the default values [84]. The synthetic map of each tubulin gene duplicated segment was generated by CIRCOS software (http://circos.ca/, access date: 20 October 2020) [85]. The putative duplicated genes were linked by connection lines. The syntenic analysis maps were constructed by using our own coded python program to prove the homologous relationship of orthologous Tubulin genes obtained from cassava and other selected species [86]. Non-synonymous (ka) and synonymous (ks) substitution of each duplicated tubulin genes were calculated using KaKs_Calculator 2.0 (https://sourceforge.net/projects/kakscalculator2/, access date: 18 November 2020) [87].

### 4.4. Phylogenetic Analysis of Cassava Tubulin Genes Family

The whole protein sequences of MeTubulin, AtTubulin, HbTubulin, ViTubulin, DrTubulin, and OsTubulin were according to the descriptions in the relevant literature and obtained from the EnsemblPlant database (http://plants.ensembl.org/, access date: 22 June 2020). All tubulin proteins were compared by ClustalW [88]. The maximum likelihood (ML) phylogenetic tree based on the full length of protein sequences was constructed by using MEGA 7.0, with the following preferences (Analysis: Phylogeny Reconstruction; Statistical Method: Maximum Likelihood; Test of Phylogeny: Bootstrap method; No. of Bootstrap Replications: 1000; Substitutions Type: Amino acid; Model/Method: Poisson model; Site Coverage Cutoff (%): 50; ML Heuristic Method: Nearest-Neighbor–Interchange (NNI); Initial Tree for ML: Make initial tree automatically (Default-NJ/BioNJ); Branch Swap Filter: Moderate) [89,90].

### 4.5. RNA-Seq Data Analysis of Tubulin Genes

The RNA-seq data of cassava (Appendix A) were obtained from the GEO database (https://www.ncbi.nlm.nih.gov/geo/query/acc.cgi?acc=GSE82279 access date: 22 October 2020) to research the expression profiles of *tubulin* genes in different tissue types. Based on Hiplot tool (https://hiplot.com.cn/basic/heatmap access date: 15 February 2021), the absolute FPKM (Fragments per kilobase of exon per million fragments mapped) values were divided by the average of all values, and the ratios were transformed by log2 to obtain data that are suitable for cluster displays, and the heatmap was generated [91].

### 4.6. Plant Materials and Treatments

SC8 (South China 8), a typical cassava cultivar was derived from the Tropical Crops Genetic Resource Institute (TCGRI, Danzhou, China). A large number of plants regenerated from tissue culture were obtained by using lateral buds of cassava on MS medium. Then, they were transferred to soil culture for two months to select seedlings with similar growth status for subsequent experiments. To study the expression pattern of the *FtsZ2-1* gene under different abiotic stresses and hormone treatments, further qRT-PCR analysis was carried out. For hormone treatments, cassava seedlings were sprayed with 100 μM IAA, MeJA, ABA, and GA3, respectively, and the leaves were collected at 2, 4, 8, 12, 24, and 48 h after treatment. In addition, 300 mM NaCl and 20%PEG were used to simulate salt stress and drought stress, which were consistent with the way of hormone treatments. All treated leaves were immediately frozen in liquid nitrogen and stored at −80 °C for subsequent analysis.

### 4.7. RNA Extraction and qRT-PCR Analysis

The RNA of each sample was extracted according to Trizol Reagent (Invitrogen #15596026). All RNA was analyzed by 1% agarose gel electrophoresis and then quantified with a Nanodrop ND-1000 spectrophotometer. RNA was used for the synthesis of the first strand of cDNA by using HiScript^®^ Ⅲ SuperMix for qPCR (+ gDNA wiper) Kit (Vazyme #R323) according to the manufacturer’s recommendations. ChamQTM Universal SYBR^®^ qPCR Master Mix (Vazyme #Q711) was used for qRT-PCR on a Roche Lightcyler^®^ 480. The reaction system is as follows: 5.0 μL 2 × ChamQ Universal SYBR ^®^qPCR Master Mix, 0.2 μL forward primers, 0.2 μL reverse primers, 1.0 μL template cDNA, 3.6 μL ddH_2_O. The PCR reaction was carried out under the following condition: preincubation (95 °C for 60 s), 40 cycles of three-step amplification (95 °C for 10 s, 60 °C for 15 s, and 72 °C for 15 s), melting (95 °C for 10 s, 65 °C for 15 s, and 97 °C for 1 s), and cooling (37 °C for 30 s). According to the related literature, the cassava *β-tubulin* gene (*MeTubulin1*) was suitable to be used as an internal reference for all the qRT-PCR analyses [92]. The specific primers were designed on the basis of *β-tubulin* and *FtsZ2-1* (*MeTubulin10*) CDS sequences (Appendix A) by Primer 5.0 software (*β-tubulin*-F: GTTATCCCCTTCCTCCCTCGTCT, *β-tubulin*-R: TCCTTGGTGCTCATCTCTTCC3 and *FtsZ2-1*-F: GCCATCCTCATCATTTACCGA, *FtsZ2-1*-R: TGGACATCCTAGCAAAGCAGA). Each sample was performed with three independent replications. Relative expression levels were calculated by the 2^−ΔΔCt^ method [93]. The expression of the *FtsZ2-1* gene at each time point was compared to the corresponding NTC (no treatment control). Statistical differences were analyzed by one-way ANOVA, followed by the post hoc tests, and a *p*-value less than 0.05 was deemed as significant. The data were processed and plotted in pictures by the GraphPad Prism 7.0.

## 5. Conclusions

In the study, a genome-wide analysis of the tubulin family in cassava was carried out, and 23 *MeTubulin* genes were identified. The biochemical characteristics of proteins, prediction of subcellular localization, gene structure, conservative motifs, chromosome location, and gene replication were analyzed, and their basic classification and evolutionary characteristics were established. This biological information provides plentiful resources for the functional identification of *tubulin* genes. The expression atlas analysis of the *MeTubulin* genes provides evidence of the potential functions of these genes in specific tissues. Moreover, analysis of *FtsZ2-1* gene expression under different treatments showed that the *FtsZ2-1* gene had a comprehensive response to IAA, MeJA, ABA, NaCl, and PEG. In general, this study laid a foundation for further analysis of the function of the *MeTubulin* gene family.

## Figures and Tables

**Figure 1 plants-10-00668-f001:**
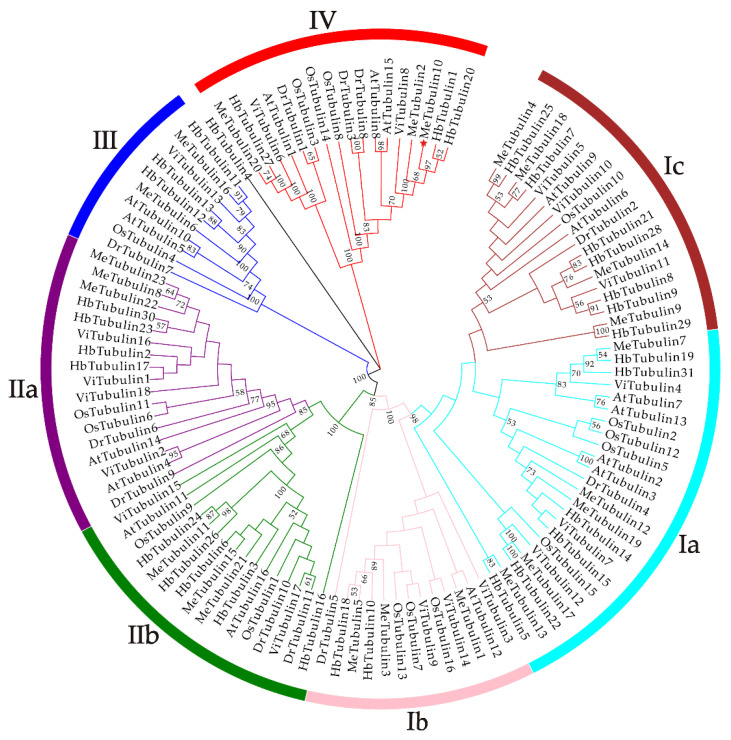
The phylogenetic tree represents the relationship between tubulin proteins in six species. The bootstrap values of less than 50% were hidden. The different-colored arcs and roman numerals indicate different groups (or subgroups) of tubulin proteins. The red star (MeTubulin10) represent FtsZ2-1 from cassava. The branch length values are shown in Appendix A.

**Figure 2 plants-10-00668-f002:**
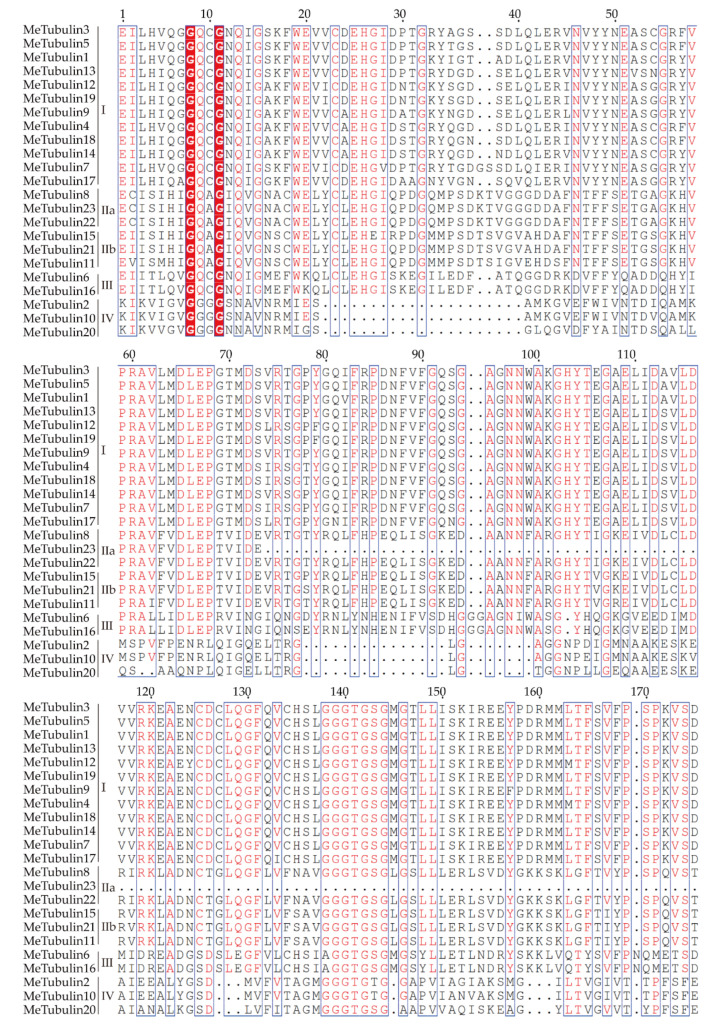
Tubulin domain sequence alignment of MeTubulin proteins. Dots represent gaps, blue frames represent the conserved region, red columns indicate the identical residues, and red letters indicate conserved residues.

**Figure 3 plants-10-00668-f003:**
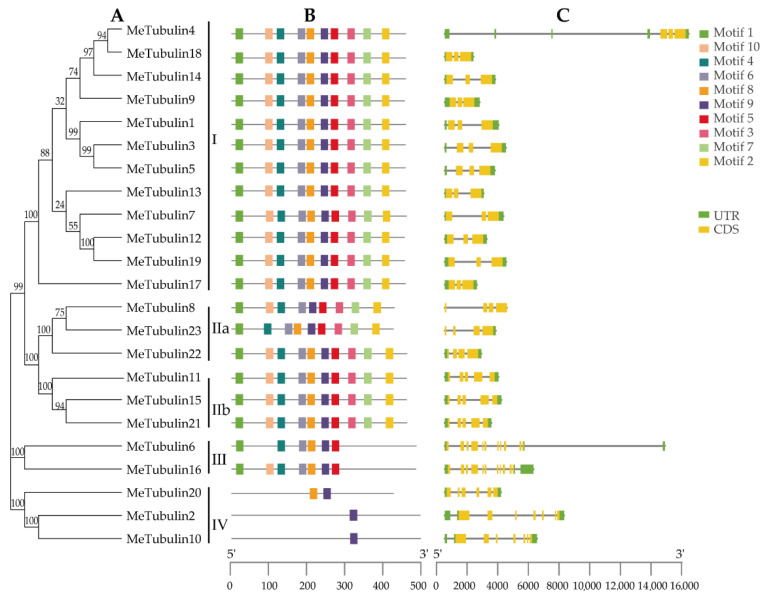
Phylogenetic relationships, gene structure, and conserved motifs in *tubulin* genes from cassava. (**A**) The phylogenetic tree was constructed according to the full-length sequences of cassava tubulin using MEGA 7.0. **(B**) The motif composition of cassava tubulin. The motifs (numbers 1–10) are represented by rectangles of different colors. Appendix A contains the amino acid sequence, E-value, and functional annotation of each motif. (**C**) Exon–intron organization of *MeTubulins*. Green rectangles represent untranslated 5′- and 3′-regions; yellow rectangle represents coding sequences (CDS); introns are represented by black lines. The scale at the bottom is to estimate the length of the protein.

**Figure 4 plants-10-00668-f004:**
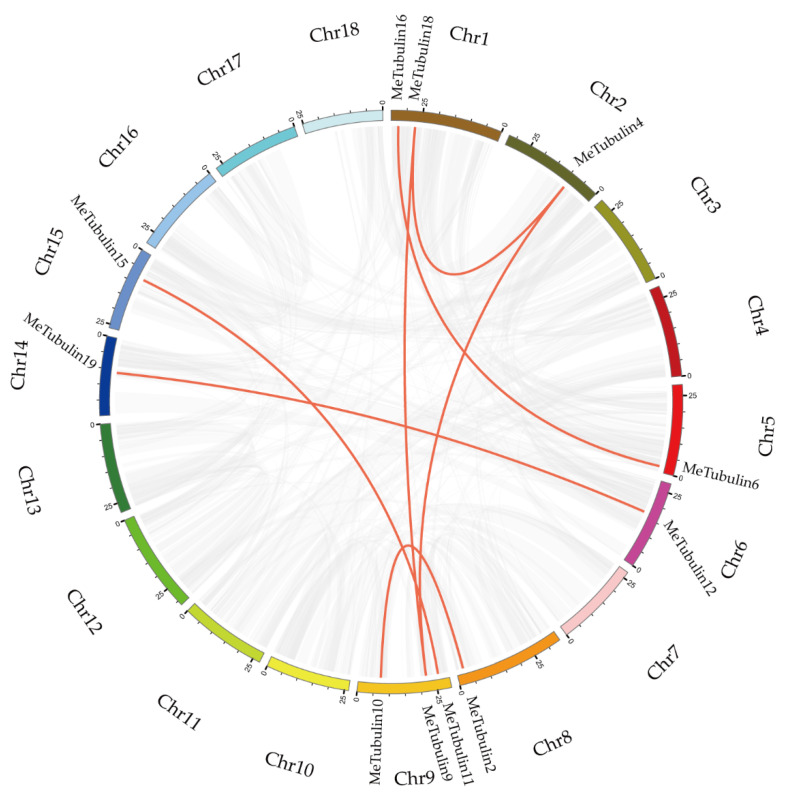
Schematic representations for the chromosomal distribution and inter-chromosomal relationships of *MeTubulins*. The synteny blocks in the cassava genome are represented by gray lines, and the duplicated *tubulin* gene pairs are by red lines. The bottom of each chromosome shows the chromosome number.

**Figure 5 plants-10-00668-f005:**
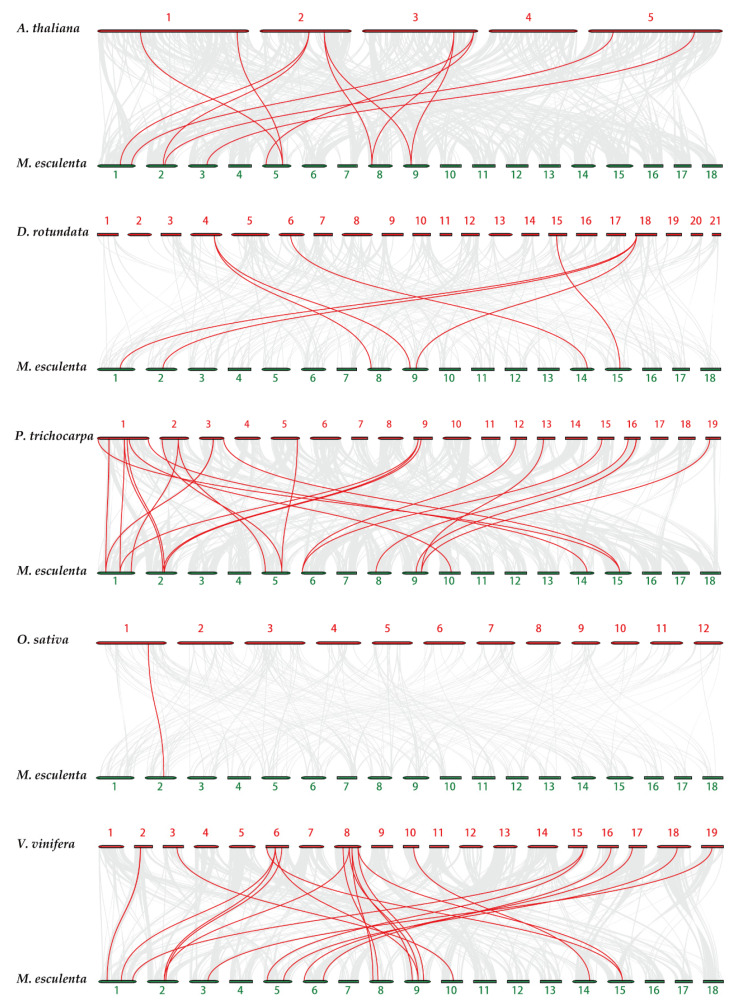
Synteny analysis of *tubulin* genes between cassava and five representative plant species. Gray lines in the background represent the collinear blocks within cassava and other plant genomes, while the red lines highlight the syntenic *tubulin* gene pairs. The species names with the prefixes ‘*M. esculenta*’, ‘*A. thaliana*’, ‘*D. rotundata*’, ‘*P. trichocarpa*’, ‘*O. sativa*’ and ‘*V. vinifera*’ indicate *Manihot esculenta*, *Arabidopsis thaliana*, *Dioscorea rotundata*, *Populus trichocarpa*, *Oryza sativa*, and *Vitis vinifera*, respectively.

**Figure 6 plants-10-00668-f006:**
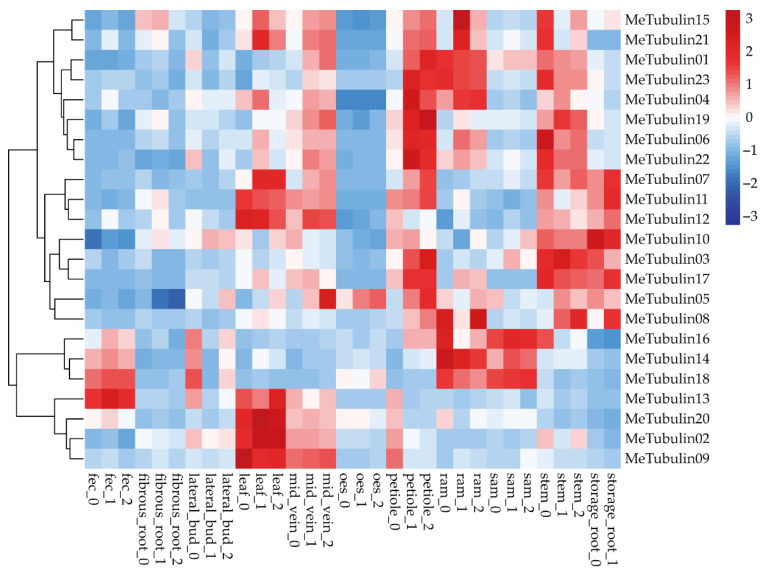
Expression profiles of the cassava *tubulin* genes. Hierarchical clustering of expression profiles of cassava *tubulin* genes in various tissue types.

**Figure 7 plants-10-00668-f007:**
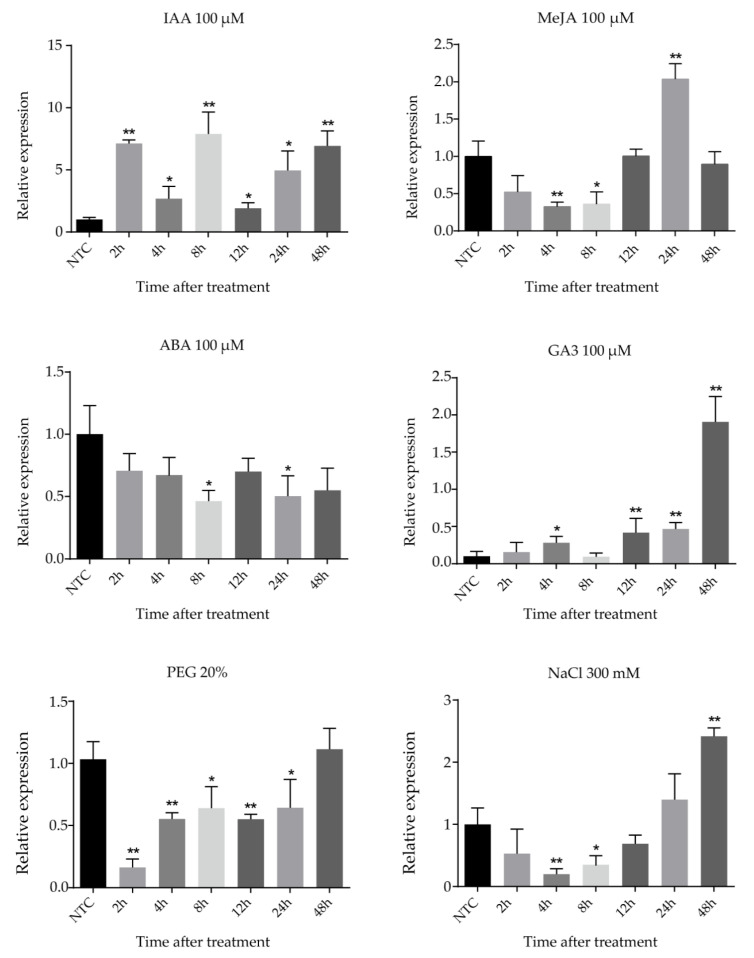
Expression profiles of *FtsZ2-1* gene in response to various stress treatments. Data were normalized to the *tubulin* gene and vertical bars indicate standard deviation. Asterisk (* significant and ** highly significant) denote significant variation (*p* < 0.05).

**Table 1 plants-10-00668-t001:** Information of 23 *MeTubulin* genes and proteins identified.

Gene Name	Gene ID	*p*-Value	Size (aa)	MW (KDa)	pI	Predicted Subcellular Location	Chr.
*MeTubulin1*	Manes.08G061700	5.6 × 10^−114^	454	50.31	4.76	cytoplasm	8
*MeTubulin2*	Manes.08G024800	2.6 × 10^−43^	492	51.13	5.56	chloroplast thylakoid membrane	8
*MeTubulin3*	Manes.02G136200	3.4 × 10^−114^	453	50.06	4.77	cytoplasm	2
*MeTubulin4*	Manes.02G123600	2.8 × 10^−114^	454	50.35	4.75	cytoplasm	2
*MeTubulin5*	Manes.05G200200	2.7 × 10^−114^	453	50.08	4.77	cytoplasm	5
*MeTubulin6*	Manes.05G025900	6 × 10^−94^	481	53.40	5.70	cytoplasm	5
*MeTubulin7*	Manes.05G147000	1.5 × 10^−112^	456	50.38	4.74	cytoplasm	5
*MeTubulin8*	Manes.13G106300	8.6 × 10^−94^	424	45.79	4.93	cytoplasm	13
*MeTubulin9*	Manes.09G100400	2.2 × 10^−113^	451	49.93	4.85	cytoplasm	9
*MeTubulin10*	Manes.09G055800	1.6 × 10^−42^	493	51.31	5.96	chloroplast stroma	9
*MeTubulin11*	Manes.09G140200	1.5 × 10^−98^	456	49.69	5.00	cytoplasm	9
*MeTubulin12*	Manes.06G058600	8 × 10^−112^	451	49.89	4.78	cytoplasm	6
*MeTubulin13*	Manes.06G007200	2 × 10^−114^	454	50.39	4.72	cytoplasm	6
*MeTubulin14*	Manes.06G147900	9 × 10^−115^	454	50.33	4.75	cytoplasm	6
*MeTubulin15*	Manes.15G108500	3.3 × 10^−99^	456	49.70	5.00	cytoplasm	15
*MeTubulin16*	Manes.01G249000	7.2 × 10^−94^	480	53.24	5.67	cytoplasm	1
*MeTubulin17*	Manes.01G061400	4.2 × 10^−112^	453	50.12	4.68	cytoplasm	1
*MeTubulin18*	Manes.01G166100	2.6 × 10^−114^	454	50.35	4.74	cytoplasm	1
*MeTubulin19*	Manes.14G124600	1.1 × 10^−113^	451	49.84	4.78	cytoplasm	14
*MeTubulin20*	Manes.03G135500	6.1 × 10^−38^	421	43.29	6.93	chloroplast stroma	3
*MeTubulin21*	Manes.03G098100	2.8 × 10^−100^	457	49.74	4.90	cytoplasm	3
*MeTubulin22*	Manes.10G087300	3.7 × 10^−100^	457	49.57	4.93	cytoplasm	10
*MeTubulin23*	Manes.10G087200	3.8 × 10^−73^	421	45.61	4.74	cytoplasm	10

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
