# Peer review of "Genome-Wide Analysis of Tubulin Gene Family in Cassava and Expression of Family Member FtsZ2-1 during Various Stress"

_plants, 2021, doi:10.3390/plants10040668_

Round 1

Reviewer 1 Report

The article entitled „Genome-Wide Analysis of Tubulin Gene Family in cassava and Transcriptional Regulation of Family Member FtsZ2-1 during Various Stress“  is analysing the group of conserved GTP-binding proteins naming Tubulins and FtsZs. In cassava 23 Tubulins are categorized in 4 groups and the study wanted to analyse the expression in different tissues and treatments with phytohormones and NaCl and PEG stress.

Abstract and Introduction

  1. The abstract is lacking the results part to show the relevance of the publication and the main findings filling a gap in the state of the art.
  2. The connection of Tubulins and FtZ is not mentioned in the Introduction. It is mentioned that FtsZ is important for division of bacteria and chloroplasts but not what Tubulin and FtsZ have in common of why this group was now analysed in general.
  3. There is no connection of stress (abiotic and hormone) to cassava and the protein group existing in the introduction.
  4. The introduction ist o short and is only mentioning three paragraphs not connected or telling a story tot he research project. Also the state of the art of previous studies or knowledge is not clearly shown to highlight the need for their study results.

Results

  1. The identification of 23 Tubulins within the genome should be validated by a table showing the p-values and domain structures of the 23 Tubulins.
  2. The subcellular localization is based on what analysis? GO with cellular composition?
  3. The phylogenetic tree of 6 species is not clearly described. Why are only these species selected and from the the group labels of I to IV are coming? From al other species and they are related to what criteria. (This could be added in the introduction already).
  4. When is the branch length okay to create a subgroup? The groups a to c have to be clarified and explained on which criteria a split was possible and is this related to the tree creation model (Maximum likelihood, Bayesian Interference) and reproducible and via bootstrap values verified.
  5. The motifs found by MEME have to be validated if the EM algorithm and the identified motifs are specific or could be also found in shuffled sequences or a set of known non tubulins.
  6. The conserved structural domains are not shown or compared based on tools like PFAM. MEME is giving motifs but not independent structural domains (super secondary elements).

Discussion

  1. The comparison of three dicotyledons and two monocotyledons is not sufficient to make statements to the phylogenetic relationships. Are these the only species with experimentally proven tubulins? If not the authors should use the genome information of different plants to identify Tubulin and tubulin like members to get a whole picture of the evolution of tubulins in Viridiplantae. Otherwise this kind of statements has to be rephrased.
  2. The statement of getting information about the functionality of the tubulins based on microarray expression data in tissues and under stress conditions is too strong. There are missing experiments to prove such statements and even the restriction to only tubulins without correlation and co-expression to known enzymes is not possible.

Material and Methods

  1. Usage of Tools like CDD, SMART and ExPASy with their parameters are not mentioned and have to be explained for reproducibility and understanding their approach.
  2. For MEME no shuffling cross-validation and other approaches for error correction are done. Also the number of motifs was fixed to 10 and the motif length is too wide so there is a overfitting to longer motifs in this approach.
  3. The creation of the phylogenetic tree without given bootstrap values but mentioned bootstraps is confusing. SO no branch is in the tree supported by a high bootstrap value?
  4. Also the trees are visualized like dendrograms without information about the branchlength.
  5. The definition of parameters has to be fully explained which is given via MEGA7 automatically with all parameters etc. It has to be explained why these parameters were chosen. Also choosing a NJ tree has to be explained and compared to ML and Bayesian inference trees as well as the input (protein, cDNA, genomicDNA etc.)
  6. In the Results it is written about Microarray datasets from GEO. In the Material and Methods the authors are writing about NextGenerationSequencing what is a huge difference in normalization etc. Also the heatmaps seems not to be normalized correctly (using z-score)
  7. The calculation in FPKM from raw counts is not clear to me based on what tool? Are the FPKM values extracted directly from the GEO or was there a mapping and annotation process involved which is not described.

Reviewer 2 Report

The manuscript entitled “Genome-Wide Analysis of Tubulin Gene Family in cassava and Transcriptional Regulation of Family Member FtsZ2-1 during Various Stress” by Li et al sent to Plants is focused on the characterization of important crop (cassava) and especially on important gene family like Tubulin. The manuscript will be of important interest in area of the biological science focused on cassava and abiotic stresses. The manuscript merit to be published after addressing the questions and remarks below.

Title:

I would like to remark that the title “…transcriptional regulation of family member FtsZ2-1 during various stress…” is not really appropriate. I think “…expression…” should replace transcriptional regulation. It the paper the authors just studied the expression patter of that gene during various chemical treatment but they didn’t go really in transcriptional regulation, which according to me should include promoter analysis etc…I would like to know the authors’ opinion.

Introduction:

The introduction is well written.

P2 L47- “…for its storage root is rich in starch…” this phrase should be changed, not very clear.

Results:

The results are well presented but as the authors focus on genome-wild study, some information about the cassava genome should be included (like size, chromosome number etc…)

P8 L168: Here the authors mentioned microarray data, but in table S8 is RNA-seq and so in the paper cited. So, they should change microarray with RNA-seq.

The name Metubulin(s) should be unified everywhere. Sometime I found MeTubulin or Metubulin. As well all latin names should be in italic.

Materials and methods:

This part is well presented but I have some questions:

Concerning the plant material treatment, it would be helpful to mention how many plants did the authors use in the treatments and how did the apply the chemicals, by spraying …?

P14L 366 The authors wrote “…melting cycle (37°C for 30 sec)? I am wondering what does it mean? Do they mean melting curve analysis or something else? I would like some clarification about that.

P13L323 .the should be replaced by .The

Overall the manuscript is well written and I would like to congratulate the authors for the good job.

Round 2

Reviewer 1 Report

Point1: The abstract is changed but the gain of this study to the research field and the new key findings is still not mentioned. The upregulation under specific hormones and tissues should be clearly put in the context of the knowledge and state of the art.

Point2: References for stating conformational similarity are missing. Also a conformational and / or sequence similarity is not equal to functional similarity. FOr the first sentence of functional similarity also references are missing.

Point3: The third point was missunderstood by the authors. Every plant is affected by abiotic stress. THe references and explanations are missing which abiotic and biotic stresses are known to be important for the production of cassava. SO is heat stress a problem in the regions of cassava production or drought or cold. Also the refernces for hormone stress and how this is added to the plants is not clearified and stated in the manuscript at stage. And referencs for stating significant effects have to be added.

Point4: The introduction is not clarifying why the authors analyze the cassava tubulin family in detail and make phylogenetic analysis etc. So at least a clear statements with the aims of the study and reasons for this in relation to the research field have to be added to allow a story to start.

Point5: I cannot see any supplemental material and so not say anything about the additional information. But important values and some schematic structural design should be added in the main manuscript because it is the starting point of the whole manuscript.

Point6: The tool Wolf Psort have to be mentioned in the Material and methods and explained what was used. In general it was a simple localization predictor used to check for targeting signals within the sequences. This approach has to be mentioned in the mansucript as it is to clarify what was done and to understand the results. Other tools like SingalP, TargetP etc. could have been used as well and also a meta analysis to create a consensus should have been considered.

Point7: The phylogenetic analysis is by different like the authors staste "genetic relationships" to simple and a neighbor Joining tree is not sufficient to this based on the evolutionary rules to consider. BAsed on previous results like Radchuck it is okay to make assumptions but the analysis has to be improved by Maximum-likelihood or baysian inference trees.

Point8: The shuffling of sequences once is not enough and this has to be repeated several times to make clear statements. Also the width is still to big and lead to a bias between the length of the motifs.

Point9: Supplements were not visible and the question went in the direction of performing a clear domain analysis and not a motif serch to make assumptions about functions. A domain and a motif are by definition different things and cannot be used in the same context.

Point10: The usage of six species is still not enough to make strong statements in the discussion about the monocotyledons and dicotyledons. Therefore more genomes are needed and also available. If the authors do not want to use these information they have to change the statements in a way that it fits to the dataset used. So going on the family level or stating that there is so far no clear divergnece observalbe etc. Even an ortholog search could help in this respect.

Material and methods:

Even when tools are used via websites it has to be mentioned clearly and the default paramters have to be mentioned or at least that default parameters where used. Tools having both a website and local existence have to be clearly stated which version was used.

Approaches like the MEME shuffling or phylogenetic analysis have to be explained in the methods parts in detail to allow the reproduction of the results.

Round 3

Reviewer 1 Report

Point1: A summary or conclusion is not giving a clear answer to a biological process. SO what is the outcome of the paper? Only a summary of different steps which are done is not clarifying the biological importance or the answer of the study.

Point3: HOw is hormone stress naturally occuring for cassava. THis becomes not clear and is not stated. THe abiotic stress I can understand on the field in very dry regions but then the salt is not completely clear. SO there should be a connection of the agrar conditions and the stress conditions which is not stated so far.

Point4: The response to analyze tubulin because of their importance in cassava growth is not clear in combination with the stress conditions. Are there evidences that show that under these stress conditions tubulins are affected? Also the FtsZ and choroplast regulation is not detailed expalined. What exactly is the role of FTsZ for chloroplasts and stress? It is not light or heat stress so moving of chloroplasts in leafs should be not affected. Also still the clear connection of FtsZ and tubulins is not explained (same evolutionary background, same structure, same pathway, same coregulation during stress)?

Point8: The motif search is still not changed. It is about the approach of MEME/MAST which has to be repeated like a jackknife approach with shuffled sequences from the same dataset (at least 10 times). Further, the motif search with a big range in the width leads to a clear known batch effect and bias towards long motifs. SO for this reason the motif width has to be adjsuted and split in several shorter ranges to make the results consistent.

Point10: One species more is not making a phylogenetic results stronger. Missing are outgroups, ortholofgs or similar sequences from all available plant species from Phytozyme for example and performing an ortholog search, then MSA and then a phylogenetic analysis of ML and baysian inference.
